# Einconv: Exploring Unexplored Tensor Network Decompositions for Convolutional Neural Networks

**Kohei Hayashi**
Preferred Networks
hayasick@preferred.jp

**Taiki Yamaguchi**[*]
The University of Tokyo
yamaguchi@hep-th.phys.s.u-tokyo.ac.jp

**Yohei Sugawara**
Preferred Networks
suga@preferred.jp

**Shin-ichi Maeda**
Preferred Networks
ichi@preferred.jp

## Abstract

Tensor decomposition methods are widely used for model compression and fast inference in convolutional neural networks (CNNs). Although many decompositions are conceivable, only CP decomposition and a few others have been applied in practice, and no extensive comparisons have been made between available methods. Previous studies have not determined how many decompositions are available, nor which of them is optimal. In this study, we first characterize a decomposition class specific to CNNs by adopting a flexible graphical notation. The class includes such well-known CNN modules as depthwise separable convolution layers and bottleneck layers, but also previously unknown modules with nonlinear activations. We also experimentally compare the tradeoff between prediction accuracy and time/space complexity for modules found by enumerating all possible decompositions, or by using a neural architecture search. We find some nonlinear decompositions outperform existing ones.

## 1   Introduction

Convolutional neural networks (CNNs) typically process spatial data such as images using multiple convolutional layers [Goodfellow et al., 2016]. The high performance of CNNs is often offset by their heavy demands on memory and CPU/GPU, making them problematic to deploy on edge devices such as mobile phones [Howard et al., 2017].

One straightforward approach to reducing costs is the introduction of a low-dimensional linear structure into the convolutional layers [Smith et al., 1997, Rigamonti et al., 2013, Tai et al., 2015, Kim et al., 2015, Denton et al., 2014, Lebedev et al., 2014, Wang et al., 2018]. This typically is done through tensor decomposition, which represents the convolution filter in sum-product form, reducing the number of parameters to save memory space and reduce the calculation cost for forwarding paths.

The manner in which this cost reduction is achieved depends heavily on the structure of the tensor decomposition. For example, if the target is a two-way tensor, i.e., a matrix, the only meaningful decomposition is $\mathbf{X} = \mathbf{UV}$, because others such as $\mathbf{X} = \mathbf{ABC}$ are reduced to that form but have more parameters. However, for higher-order tensors, there are many possible ways to perform tensor decomposition, of which only a few have been actively studied (e.g., see [Kolda and Bader, 2009]). Such multi-purpose decompositions have been applied to CNNs but are not necessarily optimal for them, because of tradeoffs between prediction accuracy and time/space complexity. The need to

---

[*]This work was completed during an internship at Preferred Networks.

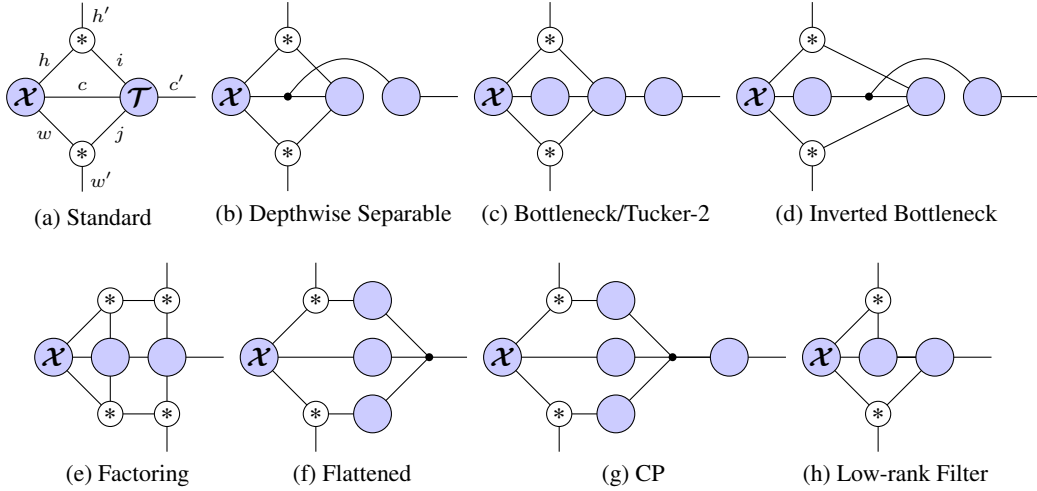

(a) Standard  (b) Depthwise Separable  (c) Bottleneck/Tucker-2  (d) Inverted Bottleneck

(e) Factoring  (f) Flattened  (g) CP  (h) Low-rank Filter

Figure 1: Visualizing linear structures in various convolutional layers, where $\mathcal{X}$ is input and $\mathcal{T}$ is a convolution kernel. The "legs" $h', w', c'$ respectively represent the spatial height, spatial width, and output channels. We will further explain these diagrams in Section 3.

consider many factors, including application domains, tasks, entire CNN architectures, and hardware limitations, makes the emergence of new optimization techniques inevitable.

In this study, we investigate a hidden realm of tensor decompositions to identify maximally resource-efficient convolutional layers. We first characterize a decomposition class specific to CNNs by adopting a flexible hypergraphical notation based on tensor networks [Penrose, 1971]. The class can deal with nonlinear activations, and includes modern light-weight CNN layers such as the bottleneck layers used in ResNet [He et al., 2015], the depthwise separable layers used in Mobilenet V1 [Howard et al., 2017], the inverted bottleneck layers used in Mobilenet V2 [Sandler et al., 2018], and others, as shown in Figure 1. The notation permits us to handle convolutions in three or more dimensions straightforwardly. In our experiments, we study the accuracy/complexity tradeoff by enumerating all possible decompositions for 2D and 3D image data sets. Furthermore, we evaluate nonlinear extensions by combining neural architecture search with the LeNet and ResNet architectures. The code implemented in Chainer [Tokui et al., 2019] is available at https://github.com/pfnet-research/einconv.

**Notation** We use the notation $[n] = \{1, \ldots, n\}$, where $n$ is a positive integer. Lower-case letters denote scalars when in ordinary type, vectors when in bold (e.g., $a, \mathbf{a}$). Upper-case letters denote matrices when in bold, tensors when in bold script (e.g. $\mathbf{A}, \mathcal{A}$).

## 2 Preliminaries

### 2.1 Convolution in Neural Networks

Consider a 2D image of height $H \in \mathbb{N}$, width $W \in \mathbb{N}$, and number of channels $C \in \mathbb{N}$, where a channel is a feature (e.g., R, G, or B) possessed by each pixel. The image can be represented by a three-way tensor $\mathcal{X} \in \mathbb{R}^{H \times W \times C}$. Typically, the convolution operation, applied to such a tensor, will change the size and the number of channels. We assume that the size of the convolution filter is odd. Let $I, J \in \{1, 3, 5, \ldots\}$ be the filter's height and width, $P \in \mathbb{N}$ be the padding size, and $S \in \mathbb{N}$ be the stride. Then, the output has height $H' = (H + 2P - I)/S + 1$ and width $W' = (W + 2P - J)/S + 1$. When we set the number of output channels to $C' \in \mathbb{N}$, the convolution layer yields an output $\mathcal{Z} \in \mathbb{R}^{H' \times W' \times C'}$ in which each element is given as

$$z_{h'w'c'} = \sum_{i \in [I]} \sum_{j \in [J]} \sum_{c \in [C]} t_{ijcc'} x_{h'_i w'_j c}, \tag{1}$$

Here $\mathcal{T} \in \mathbb{R}^{I \times J \times C \times C'}$ is a weight, which is termed as the $I \times J$ kernel, and $h'_i = (h'-1)S + i - P$ and $w'_j = (w'-1)S + j - P$ are spatial indices used for convolution. For simplicity, we omit the bias parameter. There are $IJCC'$ parameters, and the time complexity of (1) is $O(IJCH'W'C')$.

Although (1) is standard, there are several important special cases used to reduce computational complexity. The case when $I = J = 1$ is called $1 \times 1$ convolution [Lin et al., 2013, Szegedy et al., 2015]; it applies a linear transformation to the channels only, and does not affect the spatial directions. Depthwise convolution [Chollet, 2016] is arguably the opposite of $1 \times 1$ convolution: it works as though the input and output channels (rather than the spatial dimensions) are one dimensional, i.e.,

$$z_{h'w'c'} = \sum_{i \in [I]} \sum_{j \in [J]} t_{ijc'} x_{h'_i w'_j c'}. \tag{2}$$

## 2.2 Tensor Decomposition in Convolution

To reduce computational complexity, Kim et al. [2015] applied Tucker-2 decomposition [Tucker, 1966] to the kernel $\mathcal{T}$, replacing the original kernel by $\mathcal{T}^{\mathrm{T2}}$, where each element is given by

$$t^{\mathrm{T2}}_{ijcc'} = \sum_{\alpha \in [A]} \sum_{\beta \in [B]} g_{ij\alpha\beta} u_{c\alpha} v_{c'\beta}. \tag{3}$$

Here $\mathcal{G} \in \mathbb{R}^{I \times J \times A \times B}, \mathbf{U} \in \mathbb{R}^{C \times A}, \mathbf{V} \in \mathbb{R}^{C' \times B}$ are new parameters, and $A, B \in \mathbb{N}$ are rank-like hyperparameters. Note that convolution with the Tucker-2 kernel $\mathcal{T}^{\mathrm{T2}}$ is equivalent to three consecutive convolutions: $1 \times 1$ convolution with kernel $\mathbf{U}$, $I \times J$ convolution with kernel $\mathcal{G}$, and $1 \times 1$ convolution with kernel $\mathbf{V}$. The hyperparameters $A$ and $B$ may be viewed as intermediate channels during the three convolutions. Hence, when $A, B$ are smaller than $C, C'$, a cost reduction is expected, because the heavy $I \times J$ convolution is now being taken with the $A, B$ channel pair instead of with $C, C'$. The reduction ratios of the number of parameters and the inference cost for the Tucker-2 decomposition compared to the original are both at least $AB/CC'$.

Similarly, several authors [Denton et al., 2014, Lebedev et al., 2014] have employed CP decomposition [Hitchcock, 1927], which reparametrizes the kernel as

$$t^{\mathrm{CP}}_{ijcc'} = \sum_{\gamma \in [\Gamma]} \tilde{u}_{i\gamma} \tilde{v}_{j\gamma} \tilde{w}_{c\gamma} \tilde{s}_{c'\gamma}, \tag{4}$$

where $\tilde{\mathbf{U}}, \tilde{\mathbf{V}}, \tilde{\mathbf{W}}, \tilde{\mathbf{S}}$ are new parameters and $\Gamma \in \mathbb{N}$ is a hyperparameter.

## 3 The Einconv Layer

We have seen that both the convolution operation (1), (2) and the decompositions of the kernel (3), (4) are given as the sum-product of tensors with many indices. Although the indices may cause expressions to appear cluttered, they play important roles. There are two classes of indices: those connected to the output shape $(h', w', c')$ and those used for summation $(i, j, c, \alpha, \beta, \gamma)$. Convolution and its decomposition are specified by how the indices interact and are distributed into tensor variables. For example, in Tucker-2 decomposition, the spatial, input channel, and output channel information $\mathcal{G}, \mathbf{U}, \mathbf{V}$ are separated through their respective indices $(i, j), c', c$. Moreover, they are joined by two-step connections: the input channel and spatial information are connected by $\alpha$, and the output channel and spatial information by $\beta$. Here, we can consider the summation indices to be paths that deliver input information to the output.

A hypergraph captures the index interaction in a clean manner. The basic idea is that tensors are distinguished only by the indices they own and we consider them as vertices. Vertices are connected if the corresponding tensors share indices to be summed. (For notational simplicity, we will often refer to a tensor by its indices alone, i.e., $\mathcal{U} = (u_{abc})_{a \in [A], b \in [B], c \in [C]}$ is equivalent to $\{a, b, c\}$.)

As an example, consider the decomposition of a kernel $\mathcal{T}$. Let the *outer indices* $\mathcal{O} = \{i, j, c, c'\}$ be the indices of the shape of $\mathcal{T}$, the *inner indices* $\mathcal{I} = (r_1, r_2, \dots)$ be the indices used for summation, and *inner dimensions* $\mathcal{R} = (R_1, R_2, \dots) \in \mathbb{R}^{|\mathcal{I}|}$ be the dimensions of $\mathcal{I}$. Assume that $M \in \mathbb{N}$ tensors are involved in the decomposition, and let $\mathcal{V} = \{v_1, \dots, v_M \mid v_m \in 2^{\mathcal{O} \cup \mathcal{I}}\}$ denote the set of these tensors, where $2^{\mathcal{A}}$ denotes the power set of a set $\mathcal{A}$. Given $\mathcal{V}$, each inner index $r \in \mathcal{I}$ defines a

hyperedge $e_r = \{v \mid r \in v$ for $v \in \mathcal{V}\}$. Let $\mathcal{E} = \{e_n \mid n \in \mathcal{O} \cup \mathcal{I}\}$ denote the set of hyperedges. For example, suppose $\mathcal{I} = \{\alpha, \beta\}$ and $\mathcal{V} = \{\{i, j, \alpha, \beta\}, \{c, \alpha\}, \{c', \beta\}\}$; then, the undirected weighted hypergraph $(\mathcal{V}, \mathcal{E}, \mathcal{R})$ is equivalent to Tucker-2 decomposition (3).

This idea is also applicable to the convolution operation by the introduction of dummy tensors that absorb the index patterns used in convolution. Recall that in (1) the special index $h'_i$ indicates which vertical elements of the kernel and the input image are coupled in the convolution. Let $\mathcal{P} \in \{0, 1\}^{H \times H' \times I}$ be a (dummy) binary tensor where each element is defined as $p_{hh'i} = 1$ if $h = h'_i$ and 0 otherwise, and let $\mathcal{Q} \in \{0, 1\}^{W \times W' \times J}$ be the horizontal counterpart of $\mathcal{P}$. Furthermore, let us modify the index sets to $\mathcal{O} = \{h', w', c'\}$ and $\mathcal{I} = (h, w, i, j, c)$, and the dimensions to $\mathcal{R} = (H, W, I, J, C)$. Then, vertices $\mathcal{V} = \{\{h, w, c\}, \{i, j, c, c'\}, \{h, h', i\}, \{w, w', j\}\}$ and hyperedges $\mathcal{E}$ that are automatically defined by $\mathcal{V}$ exactly represent the convolution operation (1), where we ensure that the tensor of $\{h, h', i\}$ is fixed by $\mathcal{P}$ and the tensor of $\{w, w', j\}$ is fixed by $\mathcal{Q}$.

The above mathematical explanation may sound too winding, but visualization will help greatly. Let us introduce several building blocks for the visualization. Let a circle (vertex) indicate a tensor, and a line (edge) connected to the circle indicate an index associated with that tensor. When an edge is connected on only one side, it corresponds to an outer index of the tensor; otherwise, it corresponds to an inner index used for summation. The summation and elimination of inner indices is called *contraction*. For example,

$$ \underset{i}{-}\mathcal{A}\underset{j}{-}\mathcal{B}\underset{k}{-} = \underset{i}{-}\mathcal{C}\underset{k}{-} \iff \sum_j a_{ij}b_{jk} = c_{ik}. \tag{5} $$

A hyperedge that is connected to more than three vertices is depicted with a black dot:

$$ \underset{i}{-}\mathcal{A}\underset{j}{\bullet}\mathcal{B}\quad\mathcal{C}\underset{k}{-} \iff \sum_j a_{ij}b_j c_{jk}. \tag{6} $$

Finally, a node with symbol "$*$" indicates a dummy tensor. In our context, this implicitly indicates that some spatial convolution is involved:

$$ \mathcal{A}\underset{h}{-}\overset{\overset{h'}{\mid}}{*}\underset{i}{-}\mathcal{B} \iff \sum_{h,i} p_{hh'i}a_h b_i \tag{7} $$

The use of a single hyperedge to represent the summed inner index is the graphical equivalent of the Einstein summation convention in tensor algebra. Inspired by this equivalence and by NumPy's `einsum` function [Wiebe, 2011], we term a hypergraphically-representable convolution layer an *Einconv* layer.

## 3.1 Examples

In Figure 1, we give several examples of hypergraphical notation. Many existing CNN modules can obviously be described as Einconv layers (but without nonlinear activation).

**Separable and Low-rank Filters** Although a kernel is usually square, i.e., $I = J$, we often take the convolution separately along the vertical and horizontal directions. In this case, the convolution operation is equivalent to the application of two filters of sizes $(I, 1)$ and $(1, J)$. This can be considered the rank-1 approximation of the $I \times J$ convolution. A separable filter [Smith et al., 1997] is a technique to speed up convolution when the filter is exactly of rank one. Rigamonti et al. [2013] extended this idea by approximating filters as low-rank matrices for a single input channel, and Tai et al. [2015] further extended it for multiple input channels (Figure 1h).

**Factored Convolution** In the case of a large filter size, factored convolution is commonly used to replace the large filter with multiple small-sized convolutions [Szegedy et al., 2016]. For example, two consecutive $3 \times 3$ convolutions are equivalent to one $5 \times 5$ convolution in which the first $3 \times 3$ filter has been enlarged by the second $3 \times 3$ filter. Interestingly, the factorization of convolution is exactly represented in Einconv by adding two additional dummy tensors.

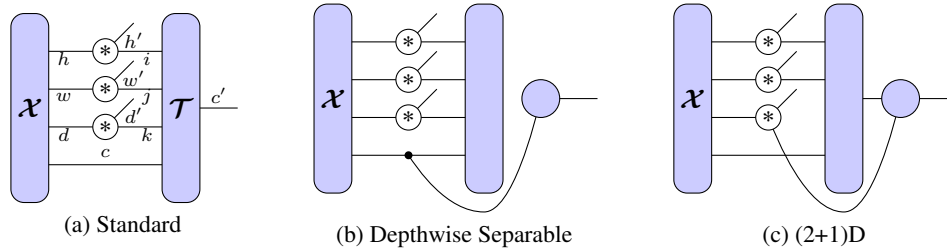

<div align="center">(a) Standard      (b) Depthwise Separable      (c) (2+1)D</div>

<div align="center">Figure 2: Graphical visualizations of 3D convolutions.</div>

**Bottleneck Layers** In ResNet [He et al., 2015], the bottleneck module is used as a building block: input channels are reduced before convolution, and then expanded afterwards. Finally, a skip connection is used, re-adding the original input. Figure 1c shows the module without the skip connection. From the diagram, we see that the linear structure of the bottleneck is equivalent to Tucker-2 decomposition.

**Depthwise Separable Convolution** Mobilenet V1 [Howard et al., 2017] is a seminal light-weight architecture. It employs depthwise separable convolution [Sifre and Mallat, 2014, Chollet, 2016] as a building block; this is a combination of depthwise and $1 \times 1$ convolution (Figure 1b), and works well with limited computational resources.

**Inverted Bottleneck Layers** Mobilenet V2 [Sandler et al., 2018], the second generation of Mobilenet, employs a building block called the inverted bottleneck module (Figure 1d). It is similar to the bottleneck module, but there are two differences. First, whereas in the bottleneck module, the number of intermediate channels is smaller than the number of input or of output channels, in the inverted bottleneck this relationship is reversed, and the intermediate channels are "ballooned". Second, there are two intermediate channels in the bottleneck module, while the inverted bottleneck has only one.

## 3.2 Higher Order Convolution

We have, thus far, considered 2D convolution, but Einconv can naturally handle higher-order convolution. For example, consider a 3D convolution. Let $d, d'$ be the input/output indices for depth, and $k$ be the index of filter depth. Then, by adding $d'$ to $\mathcal{O}$ and $d, k$ to $\mathcal{I}$, we can construct a hypergraph for 3D convolution. Figure 2 shows the hypergraphs for the the standard 3D convolution and for two light-weight convolutions: the depthwise separable convolution [Köpüklü et al., 2019], and the (2+1)D convolution [Tran et al., 2018] which factorizes a full 3D convolution into 2D and 1D convolutions.

## 3.3 Reduction and Enumeration

Although the hypergraphical notation is powerful, we need to be careful about its redundancy. For example, consider a hypergraph $(\mathcal{V}, \mathcal{E})$ where an inner index $a \in \mathcal{I}$ is only used by the $m$-th tensor, i.e, $a \in v_m$ and $a \notin v_n$ for $n \neq m$. Then, any tensors represented by $(\mathcal{V}, \mathcal{E})$, whatever their inner dimensions, are also represented by removing $a$ from every element of $\mathcal{V}$ and the $a$-th hyperedge from $\mathcal{E}$. Similarly, self loops do not increase the representability [Ye and Lim, 2018]. In terms of representability of the Einconv layer, there is no reason to choose redundant hypergraphs.[2] We therefore want to remove them efficiently.

For simplicity, let us consider the 2D convolution case, in which the results are straightforwardly extensible to higher-order cases. Let $\backslash$ denote the set difference operator and $\oslash$ denote the element-wise set difference operator, which is used to remove an index from all vertices, e.g., $\mathcal{V} \oslash a = \{v_1 \backslash a, \ldots, v_M \backslash a\}$ for index $a \in \mathcal{O} \cup \mathcal{I}$. For convenience, we define a map $\theta : \mathcal{O} \cup \mathcal{I} \to \mathbb{N}$ that

returns the dimension of an index $a \in \mathcal{O} \cup \mathcal{I}$, e.g. $\theta(i) = I$. To discuss representability, we introduce the following notation for the space of Einconv layers:

**Definition 1.** *Given vertices $\mathcal{V} = \{v_1, \ldots, v_M\}$ and inner dimensions $\mathcal{R}$, let $\mathcal{F}_\mathcal{V} : \mathbb{R}^{\times_{a \in v_1} \theta(a)} \times \cdots \times \mathbb{R}^{\times_{a \in v_M} \theta(a)} \to \mathbb{R}^{I \times J \times C \times C'}$ be the contraction of $M$ tensors along with $\mathcal{V}$. In addition, let $\mathbb{T}_\mathcal{V}(\mathcal{R}) \subseteq \mathbb{R}^{I \times J \times C \times C'}$ be the space that $\mathcal{F}_\mathcal{V}$ covers, i.e., $\mathbb{T}_\mathcal{V}(\mathcal{R}) = \{\mathcal{F}_\mathcal{V}(\mathcal{U}_1, \ldots, \mathcal{U}_M) \mid \mathcal{U}_m \in \mathbb{R}^{\times_{a \in v_m} \theta(a)}$ for $m \in [M]\}$.*

Next, we show several sufficient conditions for hypergraphs to be redundant.

**Proposition 1** (Ye and Lim 2018, Proposition 3.5). *Given inner dimensions $\mathcal{R} \in \mathbb{R}^{|\mathcal{I}|}$, if $R_a = 1$, $\mathbb{T}_\mathcal{V}(\mathcal{R})$ is equivalent to $\mathbb{T}_{\mathcal{V} \oslash a}(\ldots, R_{a-1}, R_{a+1}, \ldots)$.*

**Proposition 2.** *If $v_m \subseteq v_n$ for some $m, n \in [M]$, $\mathbb{T}_\mathcal{V}(\mathcal{R})$ is equivalent to $\mathbb{T}_{\mathcal{V} \setminus v_m}(\mathcal{R})$.*

**Proposition 3.** *If $e_a = e_b$ for $a, b \in \mathcal{I}$, $\mathbb{T}_\mathcal{V}(\mathcal{R})$ is equivalent to $\mathbb{T}_{\mathcal{V} \oslash a}(\tilde{\mathcal{R}})$ where $\tilde{\mathcal{R}} = (\ldots, R_{a-1}, R_{a+1}, \ldots, R_{b-1}, R_a R_b, R_{b+1}, \ldots)$.*

**Proposition 4.** *Assume the convolution is size-invariant, i.e., $H = H'$ and $W = W'$. Then, given filter height and width $I, J \in \{1, 3, 5, \ldots\}$, the number of possible combinations that eventually achieve $I \times J$ convolution is $\pi(\frac{I-1}{2})\pi(\frac{J-1}{2})$, where $\pi : \mathbb{N} \to \mathbb{N}$ is the partition function of integers. (See [Sloane, 2019] for examples.)*

Proposition 1 says that, if the inner dimension of an inner index is one, we can eliminate it from the hypergraph. Proposition 2 shows that, if the indices of a vertex form a subset of the indices of another vertex (e.g. $v_1 = \{a, c\}$ and $v_2 = \{a, b, c\}$), we can remove the first vertex. Proposition 3 means that a "double" hyperedge on the dimensions $A, B \in \mathbb{N}$ is reduced to a single hyperedge on the dimension $AB$. Proposition 4 tells us the possible choices of filter size. We defer the proofs to the Supplementary material. By combining the above propositions, we can obtain the following theorem:

**Theorem 1.** *If the number of inner indices and the filter size is finite, the set of nonredundant hypergraphs representing convolution (1) is finite.*

To enumerate nonredundant hypergraphs, we first use the condition of Proposition 2. Because of the vertex-subset constraint in Proposition 2, a valid vertex set must be a subset of the power set of all the indices $\mathcal{O} \cup \mathcal{I}$, and its size is at most $2^{2^{|\mathcal{O} \cup \mathcal{I}|}}$. After enumerating the vertex sets satisfying this constraint, we eliminate some of them using the other propositions.[3] We used this algorithm in the experiments (Section 6).

## 4 Nonlinear Extension

Tensor decomposition involves multiple linear operations, and each vertex can be seen as a linear layer. For example, consider a linear map $\mathbf{W} : \mathbb{R}^C \to \mathbb{R}^{C'}$. If $\mathbf{W}$ is written as a product of three matrices $\mathbf{W} = \mathbf{A}\mathbf{B}\mathbf{C}$, we can consider the linear map to be a composition of three linear layers: $\mathbf{W}(\mathbf{x}) = (\mathbf{A} \circ \mathbf{B} \circ \mathbf{C})(\mathbf{x})$ for a vector input $\mathbf{x} \in \mathbb{R}^C$. This might lead one to conclude that, in addition to reducing computational complexity, tensor decomposition with many vertices also contributes to an increase in representability. However, because the rank of $\mathbf{W}$ is determined by the minimum rank of either $\mathbf{A}, \mathbf{B}$, or $\mathbf{C}$, and the representability of a matrix is solely controlled by its rank, adding linear layers does not improve representability. This problem arises in Einconv layers.

A simple solution is to add nonlinear functions between linear layers. Although this is easy to implement, enumeration is no longer possible, because the equivalence relation becomes non-trivial with the introduction of nonlinearity, causing an infinite number of candidates to exist. It is not possible to enumerate an infinite number of candidates, thus an efficient neural architecture search algorithm ( [Zoph and Le, 2016]) is needed. Many such algorithms have been proposed, based on genetic algorithms (GAs) [Real et al., 2018], reinforcement learning [Zoph and Le, 2016], and other methods [Zoph et al., 2018, Pham et al., 2018]. In this study, we employ GA because hypergraphs have a discrete structure that is highly compatible with it. As multiobjective optimization problems need to be solved (e.g., number of parameters vs. prediction accuracy), we use the nondominated sorting

genetic algorithm II (NSGA2) [Deb et al., 2002], which is one of the most popular multiobjective GAs. In Section 6.2 we will demonstrate that we can find better Einconv layers by GA than by enumeration.

# 5 Related Work

Tensor network notation, a graphical notation for linear tensor operations, was developed by the quantum many-body physics community (see tutorial by Bridgeman and Chubb [2017]). Our notation is basically a subset of this, except that ours allows hyperedges. Such hyperedges are convenient for representing certain convolutions, such as depthwise convolution (see Figure 1b; the inclusion of the rightmost vertex indicates depthwise convolution). The reduction of redundant tensor networks was recently studied by Ye and Lim [2018], and we extended the idea to include convolution (Section 3.3).

There are several studies that combine deep neural networks and tensor networks. Stoudenmire and Schwab [2016] studied shallow fully-connected neural networks, where the weight is decomposed using the tensor train decomposition [Oseledets, 2011]. Novikov et al. [2015] took a similar approach to deep feed-forward networks, which was later extended to recurrent neural networks [He et al., 2017, Yang et al., 2017]. Cohen and Shashua [2016] addressed a CNN architecture that can be viewed as a huge tensor decomposition. They interpreted the entire forward process, including the pooling operation, as a tensor decomposition; this differs from our approach of reformulating a single convolutional layer. Another difference is their focus on a specific decomposition (hierarchical Tucker decomposition [Hackbusch and Kühn, 2009]); we do not impose any restrictions on decomposition forms.

# 6 Experiments

We examined the performance tradeoffs of Einconv layers in image classification tasks. We measured time complexity by counting the FLOPs of the entire forwarding path, and space complexity by counting the total number of parameters. All the experiments were conducted on NVIDIA P100 and V100 GPUs. The details of the training recipes are described in the Supplementary material.

## 6.1 Enumeration

First, we investigated the basic classes of Einconv for 2D and 3D convolutions. For 2D convolution with a filter size of $3 \times 3$, we enumerated the 901 nonredundant hypergraphs having at most two inner indices, where the inner dimensions were all fixed to 2. In addition to these, we compared baseline Einconv layers that include nonlinear activations and/or more inner indices. We used the Fashion-MNIST dataset [Xiao et al., 2017] to train the LeNet-5 network [LeCun et al., 1998]. The result (Figure 3) shows that, in terms of FLOPs, two baselines (standard and CP) achieve Pareto optimality, but other nameless Einconv layers fill the gap between those two.

Similarly, for a $3 \times 3 \times 3$ filter, we enumerated 3D Einconv having at most one inner index, of which there were 492 instances in total. We used the 3D MNIST dataset [de la Iglesia Castro, 2016] with architecture inspired by C3D [Tran et al., 2014]. The results (Figure 4) show that, in contrast to the 2D case, the baselines dominated the Pareto frontier. This could be because we did not enumerate the case with two inner indices due to its enormous size.[4]

## 6.2 GA Search with Non-linear Activation

Next, we evaluated the full potential of Einconv by combining it with a neural architecture search. In contrast to the previous experiments, we used Einconv layers from a larger space, i.e., we allowed nonlinear activations (ReLUs), factoring-like multiple convolutions, and changes of the inner dimensions. We employed two architectures: LeNet-5 and ResNet-50. We trained LeNet-5 with the Fashion-MNIST dataset, and the ResNet-50 with the CIFAR-10 dataset. Note that, for ResNet-50, a significant number of Einconv instances could not be trained because the GPU memory

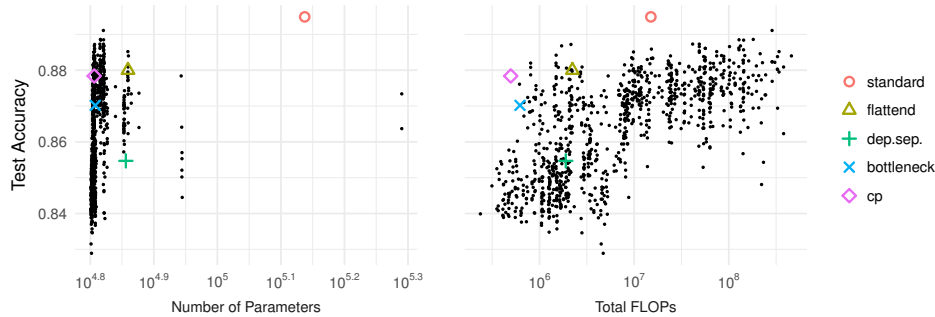

Figure 3: Enumeration of 2D Einconv for LeNet-5 trained with Fashion-MNIST. Black dots indicate unnamed tensor decompositions found by the enumeration.

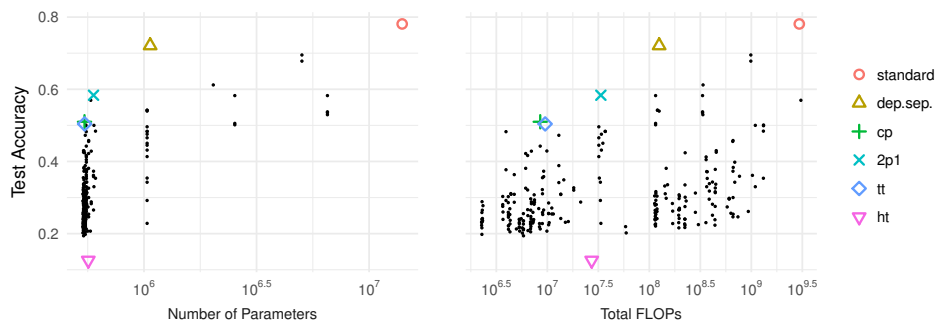

Figure 4: Enumeration of 3D Einconv for C3D-like networks trained with 3D MNIST, where `2p1`, `tt`, and `ht` mean (2+1)D convolution [Tran et al., 2018], tensor train decomposition [Oseledets, 2011], and hierarchical Tucker decomposition [Hackbusch and Kühn, 2009], respectively.

was insufficient. For the GA search, we followed the strategy of AmoebaNet [Real et al., 2018]: we did not use crossover operations, and siblings were produced only by mutation. Five mutation operations were prepared for changing the number of vertices/hyperedges and two for changing the order of contraction.[5] We set test accuracy and the number of parameters as multiobjectives to be optimized by NSGA2.

The results of LeNet-5 (Figure 5) show the tradeoff between the multiobjectives. Within the clearly defined and relatively smooth Pareto frontier, nameless Einconv layers outperform the baselines. The best accuracy achieved by Einconv was $\sim 0.92$, which was better than that of the standard convolution ($\sim 0.91$). Although the results of ResNet-50 (Figure 6) show a relatively rugged Pareto frontier, Einconv still achieves better tradeoffs than named baselines other than the standard and CP convolutions.

## 7    Conclusion and Discussion

Herein, we studied hypergraphical structures in CNNs. We found that a variety of CNN layers may be described hypergraphically, and that there exists an enormous number of variants never previously encountered. We found experimentally that the Einconv layers, the proposed generalized CNN layers, yielded excellent results.

One striking observation from the experiments is that certain existing decompositions, such as CP decomposition, consistently achieved good accuracy/complexity tradeoffs. This empirical result is somewhat unexpected; there is no theoretical reason that existing decompositions should outperform

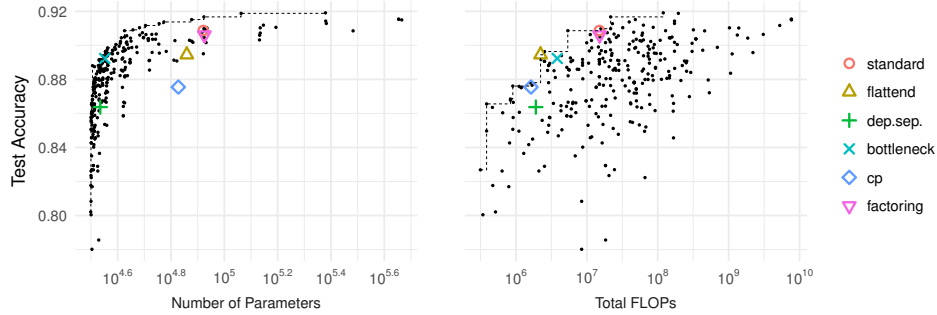

Figure 5: GA search of 2D Einconv for LeNet-5 trained with Fashion-MNIST. Black dots indicate unnamed tensor decompositions found by the GA search.

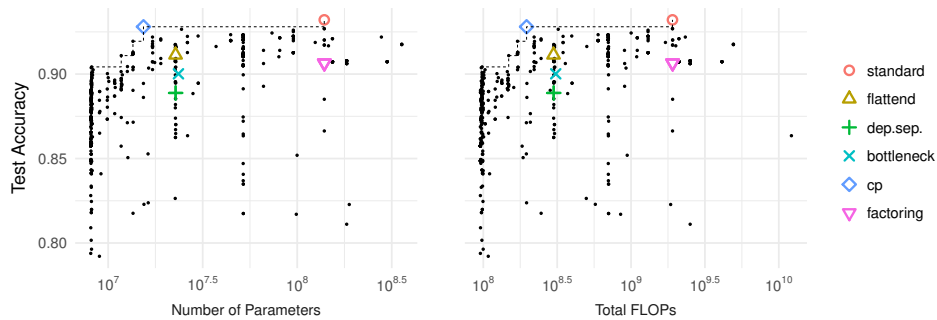

Figure 6: GA search of 2D Einconv for ResNet-50 trained with CIFAR-10.

the new, unnamed ones. Developing a theory capable of explaining this phenomenon, or at least of characterizing the necessary conditions (e.g. symmetricity of decomposition) to achieve good tradeoffs would be a promising (but challenging) direction for future work.

One major limitation at present is the computational cost of searching. For example, the GA search for ResNet-50 in Section 6.2 took 829 CPU/GPU days. This was mainly because of the long training periods (approximately 10 CPU/GPU hours for each training), but also because the GA may not have been leveraging the information on hypergraphs well. Although we incorporated some prior knowledge of hypergraphs such as the proximity regarding edge removing and vertex adding through mutation operations, simultaneous optimization of hypergraph structures and neural networks using sparse methods such as LASSO or Bayesian sparse models may be more promising.

### Acknowledgments

We thank our colleagues, especially Tommi Kerola, Mitsuru Kusumoto, Kazuki Matoya, Shotaro Sano, Gentaro Watanabe, and Toshihiko Yanase, for helpful discussion, and Takuya Akiba for implementing the prototype of an enumeration algorithm. We also thank Jacob Bridgeman for sharing an elegant TikZ style for drawing tensor network diagrams. We finally thank the anonymous (meta-)reviewers for helpful comments and discussion.

## Footnotes

[2]It might be possible that some redundant Einconv layer outperforms equivalent nonredundant ones, because parametrization influences optimization. However, we focus here on representability alone.

[3]For more details, see the real code: `https://github.com/pfnet-research/einconv/blob/master/enumerate_graph.py`

[4]For 3D convolution, the number of tensor decompositions having two inner indices is more than ten thousand. Training all of them would require 0.1 million CPU/GPU days, which was infeasible with our computational resources.

[5]See `https://github.com/pfnet-research/einconv/blob/master/mutation.py` for implementation details.

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
