[Supplementary Material]

# Appendix

## A   Proofs in Section 3.3

*Proof of Proposition 2.* Let $\odot$ denote element-wise multiplication with broadcasting[6], and $m < n$. Then the contraction of $\mathcal{U}_1, \ldots, \mathcal{U}_M$ along with $\mathcal{V}$ is written as $\mathcal{F}_\mathcal{V}(\mathcal{U}_1, \ldots) = \mathcal{F}_{\mathcal{V} \backslash v_m}(\mathcal{U}_1, \ldots, \mathcal{U}_{m-1}, \mathcal{U}_{m+1}, \ldots, \mathcal{U}_{n-1}, \mathcal{U}_m \odot \mathcal{U}_n, \ldots)$. Since $\mathcal{U}_m \odot \mathcal{U}_n \in \mathbb{R}^{\times_{a \in v_n} \theta(a)}$ for any $\mathcal{U}_m$ and $\mathcal{U}_n$, $\mathcal{F}_\mathcal{V}(\ldots)$ reduces to $\mathcal{F}_{\mathcal{V} \backslash v_m}(\ldots)$. $\qquad\square$

*Proof of Proposition 3.* Let $\mathrm{Col}(\mathcal{U}_m, a, b)$ denote the collapsing operator that reshapes tensor $\mathcal{U}_m$ by concatenating its indices $\{a, b\}$ if $\{a, b\} \subseteq v_m$, and creates a new index $b'$ where its inner dimension is $R_{b'} = R_a R_b$. Since $\mathcal{F}_\mathcal{V}(\mathcal{U}_1, \mathcal{U}_2, \ldots) = \mathcal{F}_{\mathcal{V} \otimes a}(\mathrm{Col}(\mathcal{U}_1, a, b), \mathrm{Col}(\mathcal{U}_2, a, b), \ldots)$, Proposition 3 can be proved using the same technique as in the proof of Proposition 2. $\qquad\square$

*Proof of Proposition 4.* Suppose we have $M$ size-invariant convolutions of size $(I_1, J_1), \ldots, (I_M, J_M)$. By simple calculation, we see that the final convolution size $(I, J)$ is determined by $I = 1 + \sum_{m \in [M]}(I_m - 1)$ and $J = 1 + \sum_{m \in [M]}(J_m - 1)$. Therefore, possible choices are to change $\{I_m, J_m \mid m \in [M]\}$ with varying $M >= \min(\frac{I-1}{2}, \frac{J-1}{2})$. The problem is thus reduced to the partition of integers, as stated. $\qquad\square$

*Proof of Theorem 1.* For simplicity, consider 2D convolution with a $3 \times 3$ filter ($I = J = 3$). Suppose we have $L \in \mathbb{N}$ inner indices $\mathcal{A} = \{c, r_1, \ldots, r_{L-1}\}$. According to Proposition 4, the vertical index $i$ and the horizontal index $j$ have to be used only once, and in only two possible patterns: either (i) they are used on the same vertex, or (ii) they are separated on different vertices. First, we consider case (i). Assume $v_1$ contains $\{i, j\}$ and the subset of $\mathcal{A}$, which contains $2^L$ patterns, and let $\mathcal{A}_1 = v_1 \backslash \{i, j\} \subseteq 2^\mathcal{A}$ be the selected subset. Next, consider $v_2$. To avoid the redundancy described in Proposition 2, $v_2$ must contains indices that are not contained in $v_1$, which means that the choices for $v_2$ are in $2^\mathcal{A} \backslash 2^{\mathcal{A}_1}$. Repeating this process for $v_3, v_4, \ldots$, we see that the number of patterns monotonically decreases. Moreover, the maximum length of the vertices is $L + 1$, which is achieved when $\mathcal{A}_1 = \{\varnothing\}$ and each of $v_2, \ldots, v_{L+1}$ has a single inner index. Therefore the number of nonredundant hypergraphs is finite. Case (ii) may be analyzed in the same way, except the maximum length of the vertices is $L + 2$. For the case of larger filter sizes, we can employ a similar method using Proposition 4 that ensures that the number of combination patterns of the factoring convolution will be finite. $\qquad\square$

## B   Training Recipes

### B.1   Enumeration

**2D**   The architecture is `Einconv(64)−MaxPooling−Einconv(128)−MaxPooling−FC(10)-Softmax`, where `Einconv(k)` denotes an Einconv layer with $k$ output channels and `FC(k)` denotes a fully-connected layer with $k$ output units. `Maxpooling` is performed by a factor of 2 for each spatial dimension. We trained for 50 epochs using the Adam optimizer with a batch size 16, learning rate 2E-4, and weight decay rate 1E-6.

**3D**   The architecture is `Einconv(64)-ReLU-Einconv(128)-ReLU-MaxPooling-Einconv(256)-ReLU-Einconv(256)-ReLU-MaxPooling-Einconv(512)-ReLU-Einconv(512)-ReLU-GAP-FC(512)-FC(512)-FC(10)-Softmax`, where `GAP` denotes global average pooling. We applied dropout with rate 50% to fully-connected layers except the last layer. Other settings were the same as the 2D case.

### B.2   GA Search

**LeNet-5**   The architecture is `Einconv(32)−MaxPooling−Einconv(32)−MaxPooling−FC(10)-Softmax`. We trained for at most 250 epochs using the Adam optimizer with batch size 128, learning rate 2E-4, and weight decay rate 5E-4.

**ResNet-50** We replaced all the bottleneck layers in ResNet-50 that do not rescale the spatial size by Einconv layers. We trained for at most 300 epochs using momentum SGD with batch size 32, a learning rate 0.05 that was halved every 25 epochs, and a weight decay rate 5E-4. Also, we used standard data augmentation methods: random rotation, color lightning, color flip, random expansion, and random cropping.

## Footnotes

[6]`https://docs.scipy.org/doc/numpy-1.13.0/user/basics.broadcasting.html`