[Reviews · NeurIPS 2019]

Reviewer 1



The paper shows that sum-product operation using tensors (equivalent to so-called tensor networks) can be viewed as a generalization of previously used low-rank / factorization techniques and also such techniques as depth convolution. This is quite interesting. The problem is that tensor network approximation/factorization is typically used with fine-tuning, that is when the architecture is really useful (you first do approximation using SVD-type algorithms of pretrained network, and then fine-tune). Here the layer is introduced more formally, and a huge extensive GA search is applied "on top of it", which in my opinion, does not add to the contribution. However, I think the idea is interesting, and the effort spend by the authors is huge and shows that such layers are working, but does not give any hint for larger datasets (other than CIFAR).

Reviewer 2



This paper investigates the class of tensor decompositions in CNNs and demonstrates them with hypergraphical structures. The experimental evaluation is thorough and captures well the comparison of various methods. Overall, the paper is quite easy to follow. The text is clear and the figures help clarify concepts. Appropriate motivations are clearly stated in the introduction of the paper. I believe the paper will lead some perspective on tensor and hypergraph study. However, I have some concerns below. 1. Why not make comparisons of all methods in each figure? 2. What does the black circle mean in each figure? Minor comments: - In Figures 3 and 5, some symbol markers are not clear. I think the authors should plot them after black circles. - In page 3, line 93, "in a clearn manner" should be "in a clear manner".

Reviewer 3



There are some major concerns on the paper. 1. The main theoretical results is in Sec. 3.3. However, this part is not well written, the propositions 1-4 are given without any explanations about its content. The final result in Theorem 1 is not very informative. Because it is obvious that if the inner inner indices and filter size are finite, the combinations of different tensor decompositions are finite. 2. In the experiments, especially Fig. 4 on 3D filters, the results are shown without enumerate the case with two inner indices. However, the tensor decompositions are more useful for high-order case, and tensor networks e.g., TT, is powerful for compression of filters. But these methods were not compared. 3. The papers presents a general framework for various CNNs using tensor decomposition, and the results show that standard method has best performance and CP is the Pareto optimal. What do these experiments can demonstrate for? What is the main information the authors try to convey to reader is unclear. 4. The paper is not well organized. A lot of contents are presented for the well known introduction, e.g., CNN, graphical notation of tensor operations, Einconv layer, while the innovation and contribution parts are incredibly short. 5. Experiments are not very convincing, since the authors fail to explain many important parts. For example, how to enumerate many different tensor decompositions? and why the proposed method can achieve good results? Thanks for authors' response. The paper has some interesting contributions about nice connection between some existing models and different tensor decompositions. The method in this paper is just to enumerating all possible decompositions and compare the results by using different decompositions. These results are interest to shown, but this is not a practical solution at all, since we cannot enumerating all possible tensor decompositions when we train a CNN. Therefore, I will still keep evaluation score unchanged.

[Author Response · NeurIPS 2019]

We thank all the reviewers for their dedication to reading the paper and providing helpful comments.

**R1** Thank you for your positive comments. As your suggestion regarding improvements, we are planning additional experiments of a segmentation task for 3D medical data, which will be included in the camera-ready version if they make it in time.

**R2** Thank you for telling us a typo and a suggestion to improve the figure. We will fix them. For your two concerns,
1) CP and Flattened were mistakenly excluded from Fig 5. We updated the results (see the right figure). Note that Flattened was not included in the 3D result (Fig 4) as we intended, because Flattened were not defined for 3D. Also, Factoring did not appear in Figs 3&4 because Factoring could not decompose 3x3 filter in terms of convolution (it was for 5x5 or more lager filters).

2) Each black dot in figures indicate either an hyperedge in the network diagram (see Eq. (6)) or some unnamed tensor decomposition found by enumeration/GA search.

**R4** Thank you for your questions and comments. We will clarify all of them and will revise the paper accordingly, which will enhance the quality of the paper.

> **1-a. The propositions 1-4 are given without any explanations about its content.** Each of Propositions 1–4 was collectively explained right after Proposition 4.

> **1-b. The final result in Theorem 1 is not very informative.** We respectfully disagree with this claim. If we do not consider the redundancy in terms of representation carefully, we can generate an infinite number of equivalent networks (e.g., adding many 1x1 convolutions). To prove Theorem 1, we have to formulate the definition of redundancy, etc., a part of which have not been studied. Also, Theorem 1 has an additional value that its proof in supplementary material can be used as an algorithm of enumeration (see Line 196). We will explicitly explain this in the revised manuscript.

> **2. Enumeration of 3D convolutions having at most two inner indices.** We could not train them because its size is too huge. We could enumerate them, which are 10793 decompositions in total. Training all of them requires roughly 0.1 million GPU days, which is infeasible.

> **3-a. Standard method has best performance.** This is not true for Fig 5. The accuracy of the standard convolution was 0.91, but the most accurate one achieved nearly 0.92.

> **3-b. CP performed well. What is the main information the authors try to convey to reader.** Yes, CP performed well, especially for 3D convolution. However, CP is just one of the Pareto solutions, and we have to use other solutions when the computational resources are more limited. The main message of our results is the following. The existing tensor decompositions can be Pareto optimal, but they are very sparse; However, our method can densify them.

> **4. A lot of contents are presented for the well known introduction, e.g., CNN, tensor networks, Einconv layer.** The light-weight architecture of CNN is well-known in the CNN community. Tensor network is also well-known in the tensor/physics community. However, the intersection of them – the number of people who know both – is incredibly small, we believe. Connecting the different communities and introducing the new viewpoint for light-weight CNNs is one of our main contributions, which are also recognized by other reviewers.

Note that we have first introduced the notion of Einconv layers, so it should not have been well known.

> **5-a. How to enumerate many different tensor decompositions?** As we answered above, the proof of Theorem 1 forms an algorithm for enumeration. We will publish the real code.

> **5-b. Why the proposed method can achieve good results?** Compare to the entire space of tensor decomposition, the existing tensor decompositions such as CP represent just the tip of the iceberg. So it is reasonable to think there exist better decompositions in the iceberg, which can be found by our method.

**Improvements: Comparison with other tensor network such as TT, HT etc.** Thank you for the suggestion.

We conducted additional experiments for TT and HT; please check the updated results (the above figure). Note that `{tt,ht}_relu` are the variants of having ReLU activation. Overall, both TT and HT are not better than CP (TT is close to CP, though).

[Meta-Review · NeurIPS 2019]

This paper relates sum-product tensor operations (a.k.a. tensor networks) to compressed/factorized convolutional layers in neural networks. In doing so, they formally define a new kind of layer, einconv layer, that generalizes previously proposed approaches for compressing CNNs. An extensive search over the space of possible layers is performed to compare new factorized layers with existing ones. The reviewers agree that the idea is original and well executed and that the paper has potential to be significant. One concern is that the proposed enumeration algorithm used in the experiments is not practical, which is true. Nonetheless, this paper opens the way for future research in this direction (how to efficiently search the space of einconv/factorized layers), demonstrates that other factorization models can be considered, and provides a clear picture of connections between tensor networks and existing literature on compressing convolutional layers, which are all significant. Minor comments: - A final proof reading of the paper is needed (e.g. first sentence of Section 3.2, line 207 "this would be happen", line 211 "We"->"we"...) - I may have missed it but I think the notation \mathbb{T}_V(R) (used to denote the set of rank R tensors factorized according to V) has not bee introduced. - In Proposition 2, I believe T_{V\v_m}(R) should be T_{V\v_m}(\tilde{R}) with \tilde{R} = R \ {R_i | i \in v_m}.